# Differentiable molecular simulation can learn all the parameters in a coarse-grained force field for proteins

**Joe G. Greener** *, **David T. Jones**

Department of Computer Science, University College London, London, United Kingdom

* j.greener@ucl.ac.uk

**Data Availability Statement:** The learned potential, simulation scripts and training code are made available under a permissive license at https://github.com/psipred/cgdms.

## Abstract

Finding optimal parameters for force fields used in molecular simulation is a challenging and time-consuming task, partly due to the difficulty of tuning multiple parameters at once. Automatic differentiation presents a general solution: run a simulation, obtain gradients of a loss function with respect to all the parameters, and use these to improve the force field. This approach takes advantage of the deep learning revolution whilst retaining the interpretability and efficiency of existing force fields. We demonstrate that this is possible by parameterising a simple coarse-grained force field for proteins, based on training simulations of up to 2,000 steps learning to keep the native structure stable. The learned potential matches chemical knowledge and PDB data, can fold and reproduce the dynamics of small proteins, and shows ability in protein design and model scoring applications. Problems in applying differentiable molecular simulation to all-atom models of proteins are discussed along with possible solutions and the variety of available loss functions. The learned potential, simulation scripts and training code are made available at https://github.com/psipred/cgdms.

## Introduction

Molecular simulation has been successful in making predictions and understanding experimental data [1, 2]. Treating the system with an appropriate level of complexity, usually all-atom molecular mechanics or residue-level coarse-graining when simulation proteins, is necessary to access the timescales required for the property under investigation [3, 4]. It is generally agreed that force fields are not optimally parameterised [5, 6], for example significant effort has gone into modifying a handful of parameters in standard force fields to better represent both ordered and disordered proteins [7–9]. Such efforts have improved the force fields without changing their functional form, an attractive proposition when the alternative is adding complexity that restricts the timescales available for study.

Meanwhile, deep learning has had a major impact on many areas of biology, achieving state of the art performance in fields such as protein structure prediction [10]. A number of groups have applied these advances to molecular simulations [11–13] including learning coarse-grained potentials [14–18], learning quantum mechanical potentials [19–22], improving

**Funding:** This work was supported by the European Research Council (https://erc.europa.eu) Advanced Grant "ProCovar" (project ID 695558) awarded to DTJ. The funders had no role in study design, data collection and analysis, decision to publish, or preparation of the manuscript.

**Competing interests:** The authors have declared that no competing interests exist.

sampling [23, 24], and improving atom typing [25]. Whilst promising, many of these approaches show limited success when used on systems they were not trained on. Methods trained on trajectory data also suffer from a lack of standardised simulations across many systems due to the high cost of obtaining such trajectories. Other groups have used inventive machine learning strategies for end-to-end protein structure prediction [26] and to score static conformations [27, 28].

The idea of differentiating through the numerical solution of Newton's equations of motion to obtain gradients that can be used to improve a learned force field is called differentiable molecular simulation (DMS). It has been discussed [29], but is yet to produce generally useful force fields. Conceptually the idea is related to recent work on neural differential equations [30–32]. The idea is appealing because a large number of parameters can be improved at once, rather than the small numbers currently modified. The gradients are also exact, at least with respect to the numerical integration and the loss function used. Only recently has the hardware and software been available to run such simulations. The variety of available loss functions and ability to calculate exact gradients for all parameters suggest DMS could be the next step in the steady improvement of force fields [33].

Previous studies utilising DMS have trained neural network potentials, ranging from graph neural networks [29] to ambitious protein folding simulators running Langevin dynamics [34]. Whilst training neural networks may be an effective solution, as discussed above there is much room for improvement in existing force fields. It makes sense to try and improve these where possible rather than moving to a new functional form, which has the additional advantage of retaining the physical interpretability of conventional force fields. Neural networks are also slower to run than existing force fields, meaning that to avoid reducing available simulation time the network has to be trained to jump multiple time steps, a rather different problem to learning the instantaneous potential. This work is also influenced by a number of studies that compare native and training ensembles to improve force fields for protein folding using maximum likelihood, contrastive divergence and related approaches [35–45]. These methods are able to modify many parameters at once, but they generally involve comparing conformations rather than obtaining gradients of some loss function through the simulation and are hence limited in the properties they can target.

In this study we use automatic differentiation (AD) [46], the procedure used to train neural networks where it is called backpropagation, to learn all the parameters from scratch in a simple coarse-grained force field for proteins. This learned potential matches potentials derived from chemical knowledge and Protein Data Bank (PDB) statistics, reproduces native flexibility when used in simulation, is able to fold a set of small proteins not used for training, and shows promise for protein design and model scoring applications. It adds to existing coarse-grained and statistical potentials used for simulation [47–50] and model scoring approaches [51–53]. More broadly, it points to DMS as a useful technology falling under the banner of differentiable programming [54], an expansion of the principles of deep learning to the concept of taking gradients through arbitrary algorithms and utilising the known structure of the system under study [55, 56].

## Results

### Differentiable molecular simulation

In this study a coarse-grained potential for proteins is learned in which a protein is represented by 4 point particles per residue (N, Cα, C and sidechain centroid) with no explicit solvent. The potential consists of 3 components: pairwise distance potentials (including covalent bonds), bond angle potentials, and torsion angle potentials associated with the predicted secondary

structure type of a residue. Overall there are 29,360 individual potentials and 4.1m learnable parameters. The high number of parameters indicates some redundancy due to the nature of the potentials, but also demonstrates that DMS can be used to learn a large number of parameters at once. See the methods for further details on the functional form of the potentials, how forces are calculated and how the model is trained. Proteins presented in the results do not have homologs present in the training set and single sequence secondary structure prediction is used throughout the study, so the results are not due to overtraining or direct learning of evolutionary information. Assessing the method on proteins not used for training distinguishes this approach from many approaches used to date for machine learning of molecular simulations.

During training, proteins are simulated using the velocity Verlet integrator in the NVE ensemble, i.e. with no thermostat. The starting conformation is the native structure and up to 2,000 steps are run. Due to parameters shared across each time step this can be thought of as analogous to a recurrent neural network (RNN) running on a sequence of length 2,000, as shown in Fig 1A. In particular, implementing the simulation in the AD framework PyTorch [57] allows the gradients of a given loss function with respect to each learned parameter to be calculated, allowing an optimiser to change the parameters to reduce the loss function. However, the learned parameters are those of the force field rather than the weights and biases of a standard neural network. $\log(1 + R_f)$ is used as the loss function, where $R_f$ is the root-mean-square deviation (RMSD) across all atoms in the coarse-grained model between the conformation at the end of the simulation and the native structure. At the start of learning the potentials are flat, the forces are zero and the proteins distort according to the randomised starting velocities, as shown in Fig 1B. Over the course of training, the $R_f$ values decrease as the potential learns to stabilise the proteins in the training set. Previous studies have adopted a similar approach of minimising the final RMSD but used random sampling rather than exact gradients [58]. The training and validation $R_f$ values throughout training are shown in S1 Fig in S1 File.

After training on a dataset of 2,004 diverse proteins up to 100 residues long the potentials resemble those derived from chemical knowledge and PDB statistics. Due to the coarse-grained nature of the simulation the energy values, along with other properties such as the time step and the temperature used later in the thermostat, cannot be assigned standard units. As shown in Fig 2A, covalent bond distance potentials have strong minima at the correct distance and steep barriers preventing steric clashing. The steep drops at the edges are an artifact of training and do not affect simulations or energy scoring, since these values are never occupied when using the trained potential. Bond angle potentials have minima at the true values with a few degrees of tolerance allowed either side, as shown in Fig 2B. Same residue Cα-side-chain distance potentials indicate that different rotamer conformations have been learned; Fig 2C shows that the energy minima for isoleucine agree with the two minima found in the PDB distance distributions, which correspond to different rotamers. Other cases showing different rotamer conformations include glutamic acid, glutamine, lysine, methionine and tryptophan. These can be seen in the complete set of such potentials shown in S2 Fig in S1 File. The torsion angle potentials show different preferences for residues predicted as α-helical, β-sheet and coiled, see Fig 2D, and these agree with the true Ramachandran distributions. The glycine torsion angle potentials indicate lower energy regions in the positive φ space and the proline potentials display a minimum at 0˚ for the ω angle corresponding to cis-proline (data not shown). The potentials most important for the tertiary structure are the general distance potentials. As shown in Fig 2E these match potentials of mean force (PMFs) derived from the PDB in many cases, with minima for many pairs around 6 Å driving hydrophobic packing. The steep energy barriers to steric clashing do not generally extend below 4 Å because

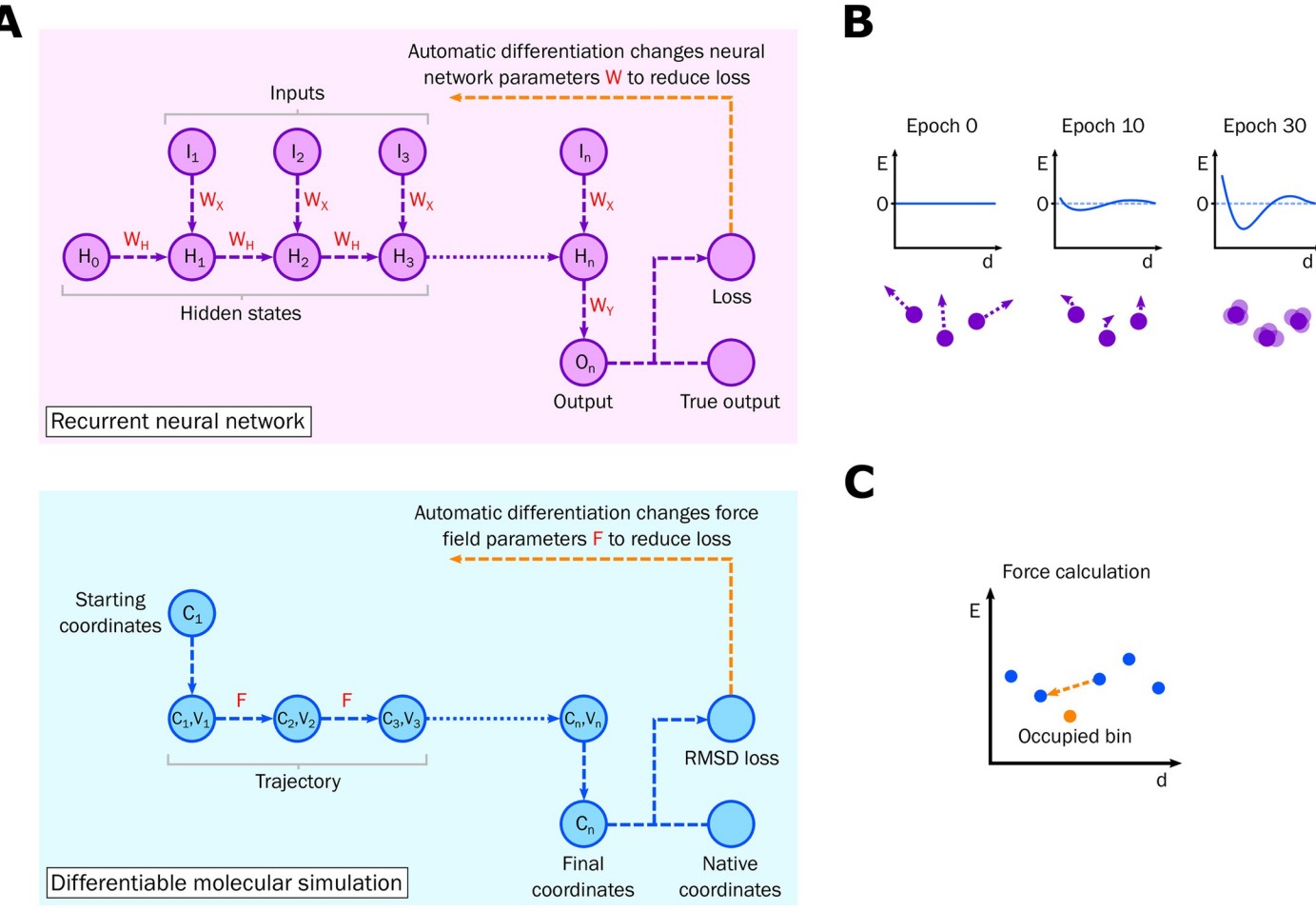

**Fig 1. Differentiable molecular simulation.** (A) The analogy between a RNN and DMS. Learnable parameters are shown in red. The same parameters are used at each step. There are many variants of RNNs; the architecture shown here has a single output for a variable length input, which could for example represent sentiment classification of a series of input words. (B) Learning the potential. A representation of a component of the potential is shown with energy $E$ plotted against inter-atomic distance $d$. At the start of training (epoch 0) the potential is flat and the atoms deform according to their starting velocities. During training the potential learns to stabilise the native structures of the training set. (C) Force calculation from the potential. Adjacent bins to the occupied bin are used to derive the force using finite differences. In this case the force acts to reduce the distance $d$. See the methods for more details.

extending these is not required to improve $R_f$ during training. Steric clashing is not seen during simulations provided that a suitable time step is used. The complete set of potentials for general sidechain-sidechain distances are shown in S3 Fig in S1 File.

Despite being trained only to minimise the final RMSD across the protein, we find that the learned potential shows local detail. The potential energy when modifying the φ and ψ torsion angles of alanine dipeptide is shown in Fig 3A. Shown in Fig 3B is the free energy calculated from an all-atom simulation in Wang et al. 2019 [15]. The learned potential matches the major low energy conformations of the all-atom model.

## Protein structure and dynamics

A learned potential can be used to run a simulation of arbitrary length since gradients are not recorded. Here we study four small, fast-folding proteins investigated with molecular simulation previously [8, 59, 60]. They all have NMR ensembles or crystal structures available from experiments [61, 62]. Details on all proteins presented in the results are in S1 Table in S1 File.

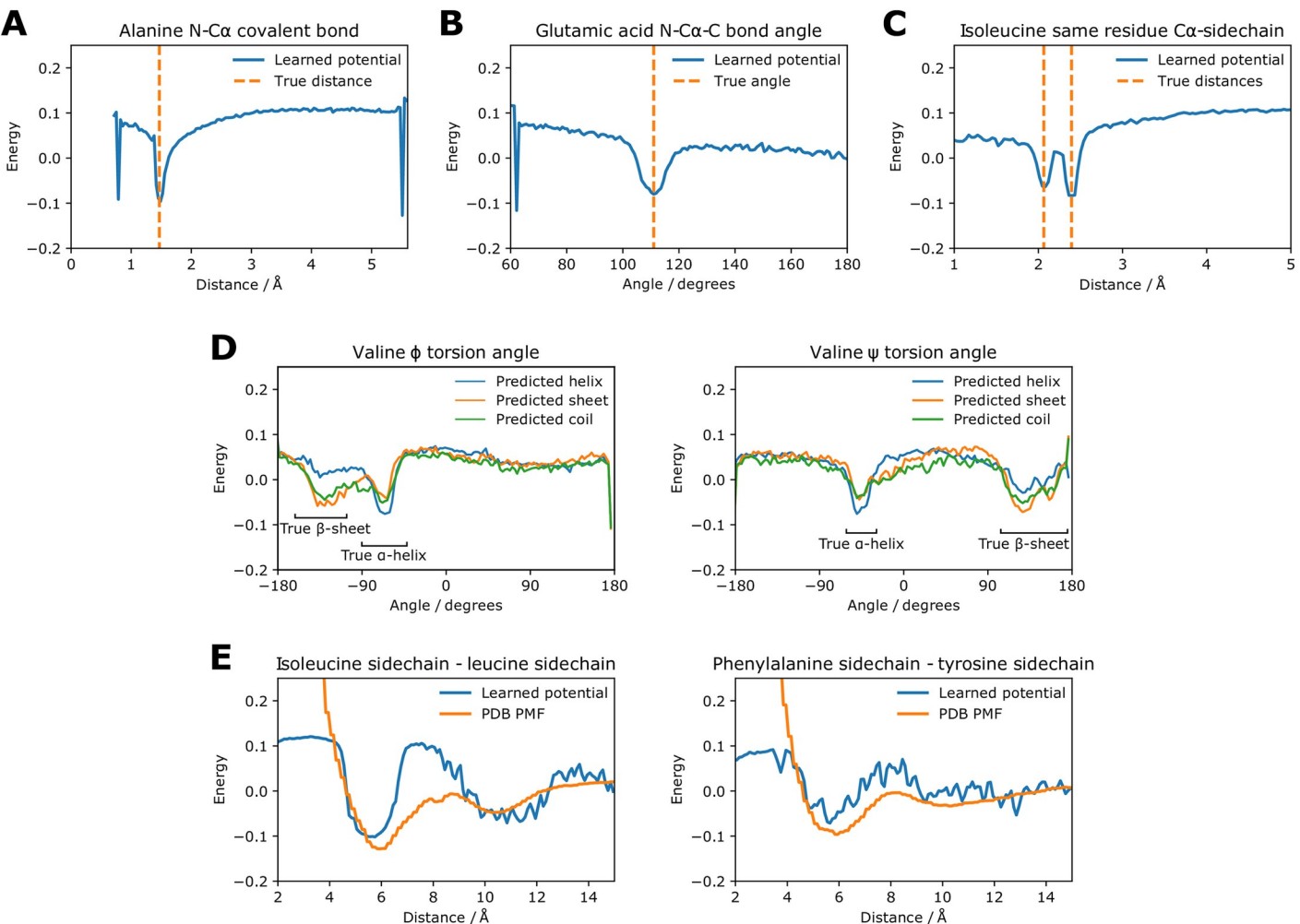

**Fig 2. Components of the learned coarse-grained potential compared to true values.** The energy scale is the same for each plot. Each individual potential consists of discrete energy values for 140 evenly-spaced bins. (A) Distance potential between N and Cα in the same alanine residue, i.e. a covalent bond. (B) Bond angle potential for the N-Cα-C angle in glutamic acid. (C) Distance potential between Cα and the sidechain centroid on the same residue in isoleucine. (D) Torsion angle potentials in valine. There are different potentials for residues predicted as α-helical, β-sheet and coiled. The true ranges of α-helices and β-sheets are shown. (E) Distance potentials between sidechains for two pairs of amino acids. A PMF calculated from the PDB is also shown for comparison.

First we investigate whether the learned potential can keep the native structures stable and reproduce residue-level flexibility. As shown in Fig 4B, the native structures are stable under simulation in the NVT ensemble using the Andersen thermostat, with Cα RMSDs less than 4 Å in all cases. Fig 3C shows that the root-mean-square fluctuation (RMSF) of the Cα atom of each residue over the simulation generally matches that of the native NMR ensembles, with an expected increase in flexibility for terminal residues. Chignolin displays more flexibility under simulation than in the NMR ensemble, likely due to the lack of explicit hydrogen bonding in the coarse-grained model to keep the β turn structure rigid. For villin HP36, we see general agreement between the residue RMSF and the crystal temperature factors.

Next, we ask whether we can fold small proteins and peptides in the NVT ensemble. We find that the (AAQAA)₃ repeat peptide folds into an α-helix over 12m steps when started from a random conformation, matching its experimental behaviour at physiological temperatures. This is accompanied by a reduction in the energy under the learned potential and is shown in

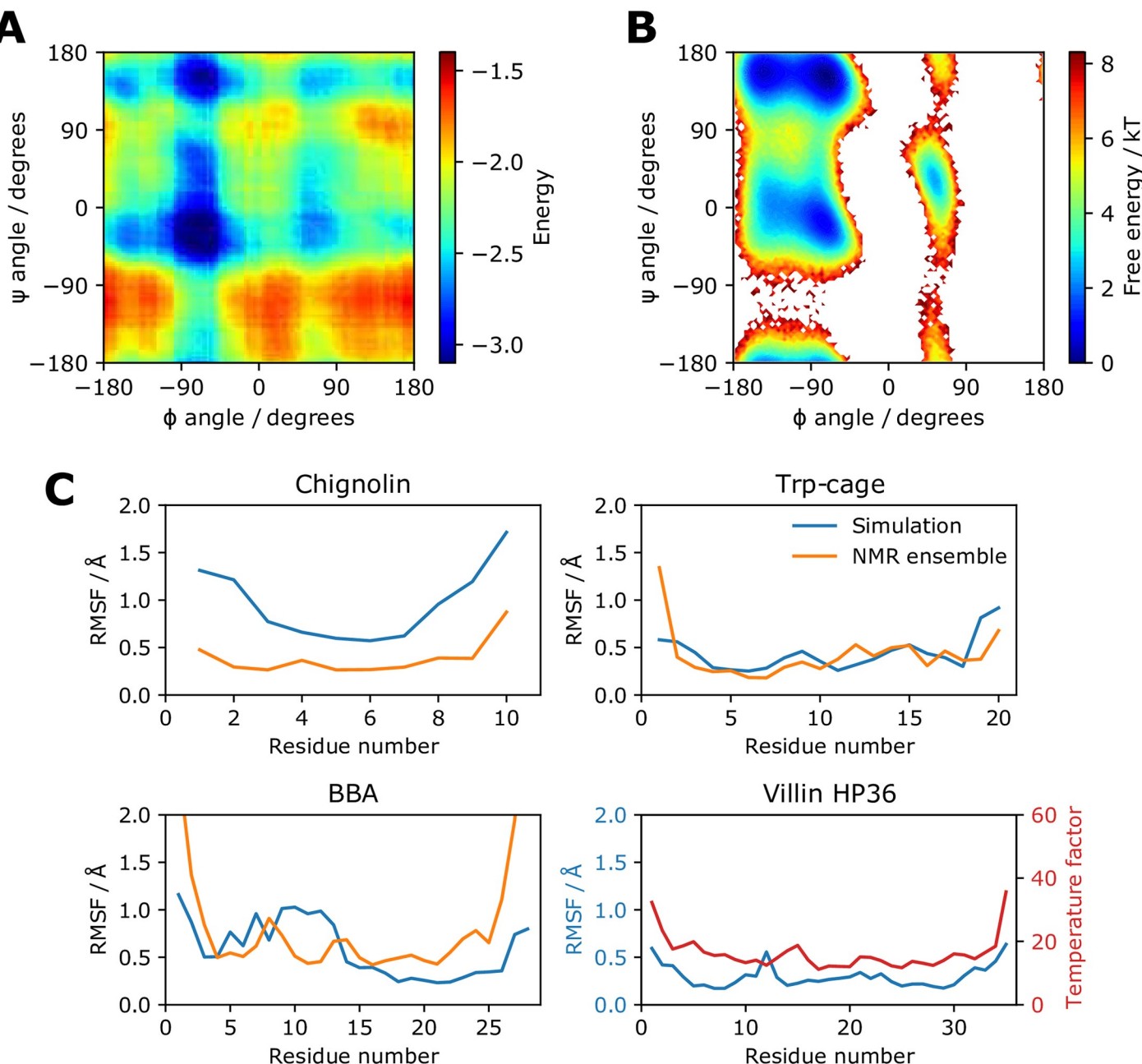

**Fig 3. Protein energy and dynamics.** (A) The energy of alanine dipeptide in the learned potential for different φ and ψ torsion angles. Conformations are generated with PeptideBuilder at intervals of 2° and scored. (B) Free energy for different φ and ψ torsion angles for all-atom alanine dipeptide. The energy is obtained from direct histogram estimation from all-atom simulations. The data is taken from Wang et al. 2019 [15] and provided by the authors. (C) Cα atom RMSF values for simulations of 6m steps using the learned potential starting from the native structure. An initial burn-in period of 6m steps was discarded. The Cα atom RMSF values from the NMR ensembles are also shown, or the Cα atom crystal temperature factors in the case of villin HP36.

Fig 4A. We do find that longer sequences are able to form the correct secondary structure, and often the correct tertiary structure given enough simulation time. However to better explore tertiary structure formation with available compute resources we started the proteins from extended conformations containing predicted secondary structure, i.e. α-helical φ/ψ angles for predicted α-helical residues and extended φ/ψ angles for residues predicted β-sheet or coiled.

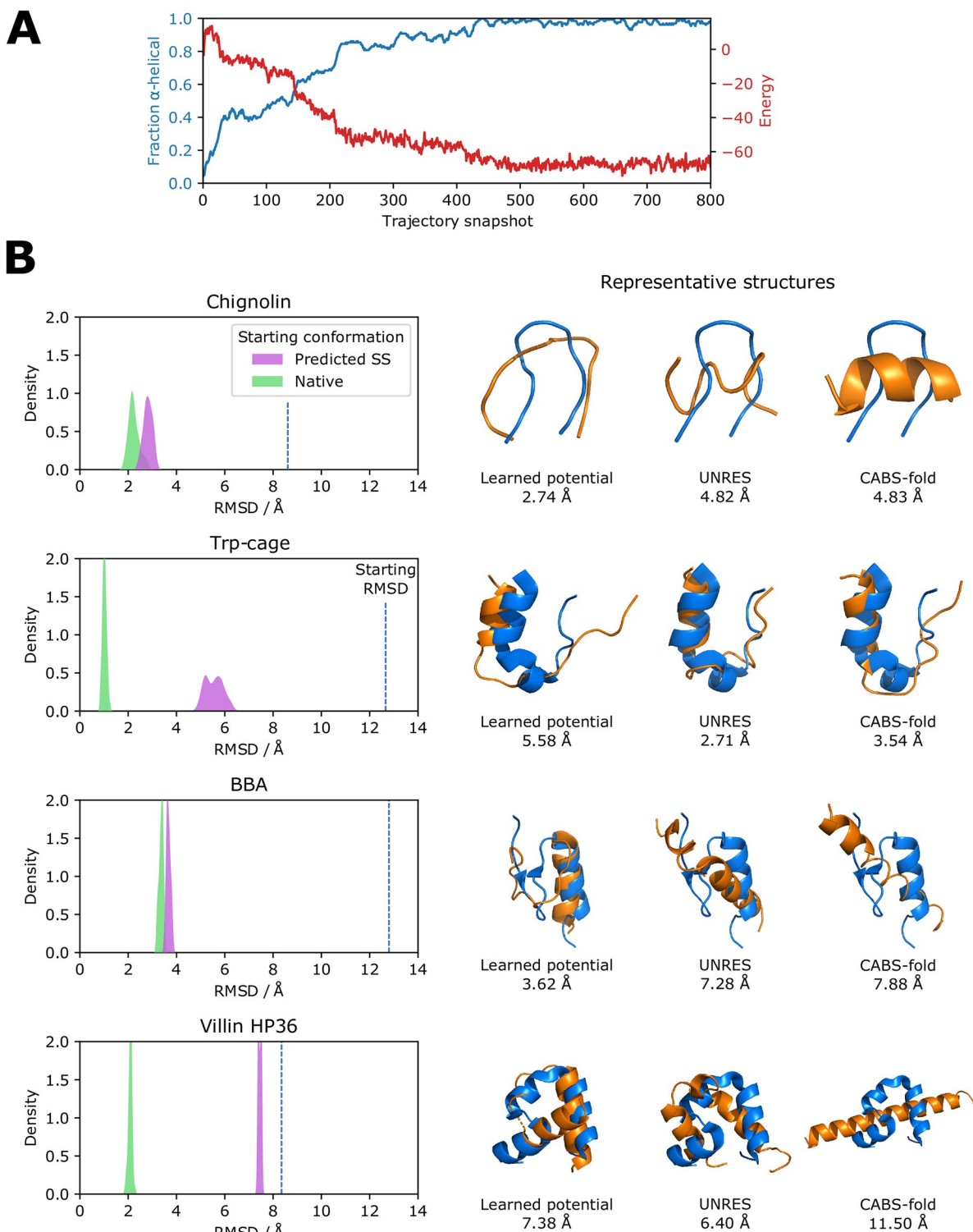

**Fig 4. Folding small proteins and peptides.** (A) The (AAQAA)$_3$ repeat peptide folds from a random starting conformation into an α-helix over 12m steps. The α-helical fraction and energy in the learned potential are shown with a snapshot taken every 15,000 steps. (B) Cα RMSD distributions from simulations of 3m steps for 4 proteins starting from predicted secondary structure and native conformations. An initial burn-in period of 9m steps was discarded. The starting Cα RMSD for the predicted secondary structure conformation is also shown. A representative structure found with MDAnalysis is shown (orange) along with models generated from the web servers of UNRES and CABS-fold. The Cα RMSD to the native structure (blue) is given for each model.

The Cα RMSD distributions of the trajectories after a burn-in period, the starting Cα RMSDs and representative models from the trajectories are shown in Fig 4B. All the proteins fold to a native-like structure over 12m steps, taking about 36 hours on a single graphics processing unit (GPU) or about 3 times as long on the central processing unit (CPU). Chignolin adopts approximately the correct β turn structure; the minimum sampled Cα RMSD of 2.15 Å is comparable to the Cα RMSD of 1.82 Å between the first model in the NMR ensemble and the crystal structure. Trp-cage lacks native helix formation in the middle of the protein but the overall shape is correct. BBA forms a native-like structure with a minimum Cα RMSD during the simulation of 3.47 Å. For villin HP36 the N-terminal helix faces the wrong direction but the location of the turns and the rest of the structure is correct. This indicates a low energy basin in which the topological mirror structure is found, a problem that can be overcome with a higher temperature (see below) or enhanced sampling.

Models are also shown from web servers implementing two popular methods that carry out coarse-grained protein folding: UNRES [47, 63], which has two interacting sites per protein, and CABS-fold [48, 64], which uses a lattice model. Both make use of predicted secondary structure, and are given the same single sequence prediction used here. Performance of the learned potential is comparable to these established methods across the proteins tested, with the learned potential able to break secondary structure elements and add turns in the correct locations. The comparison to these methods is not exact; they provide less compute resources on their servers than the 36 hours of GPU time per protein used here, whereas this method does not employ the replica exchange algorithms used to enhance sampling in UNRES and CABS-fold. Enhanced sampling approaches to predict the structures of larger proteins with the learned potential is a topic of future work. We note that larger proteins remain close to their native structure with low energy when simulated, suggesting that the native structure is in an energy minimum that can be accessed with appropriate sampling. Another point of note is that the same simulation parameters are used when simulating all four proteins (temperature 0.015, coupling constant 25). More accurate models can likely be obtained by optimising these for each protein but we did not want to risk overfitting. The (AAQAA)$_3$ repeat peptide helix formation simulations were, however, carried out at a higher temperature (temperature 0.022, coupling constant 100) to faster explore the conformational space needed to form the α-helix. We do notice some success in folding from an extended chain with these higher temperature parameters. For example, villin HP36 reaches a minimum Cα RMSD of 4.19 Å over 12m steps, with the orientation of all the helices correct.

## Protein design and model scoring

In order to see whether native sequences are optimal for native structures in the learned potential, we thread sequences with varying fractions of native amino acids onto the native backbone and calculate the energy. Since long simulations are not required for these tests we also used four more proteins investigated with similar methods previously [59, 60]. Non-native residues are drawn randomly from the background distribution of amino acids in the PDB, i.e. leucine is more likely to be chosen (9.6% chance) than tryptophan (1.2% chance). As shown in Fig 5A we find that an increased fraction of native amino acids gives a lower energy, with the native sequence lower in energy than most 90% native sequences and considerably lower in energy than less native sequences.

Next we used the potential for fixed backbone design to see if designed sequences match the native sequence. We start from a random sequence and make mutations. At each trial a residue is mutated and the mutation is accepted or rejected based on the energy change when threaded onto the native structure and the distance through the trial process (see the methods).

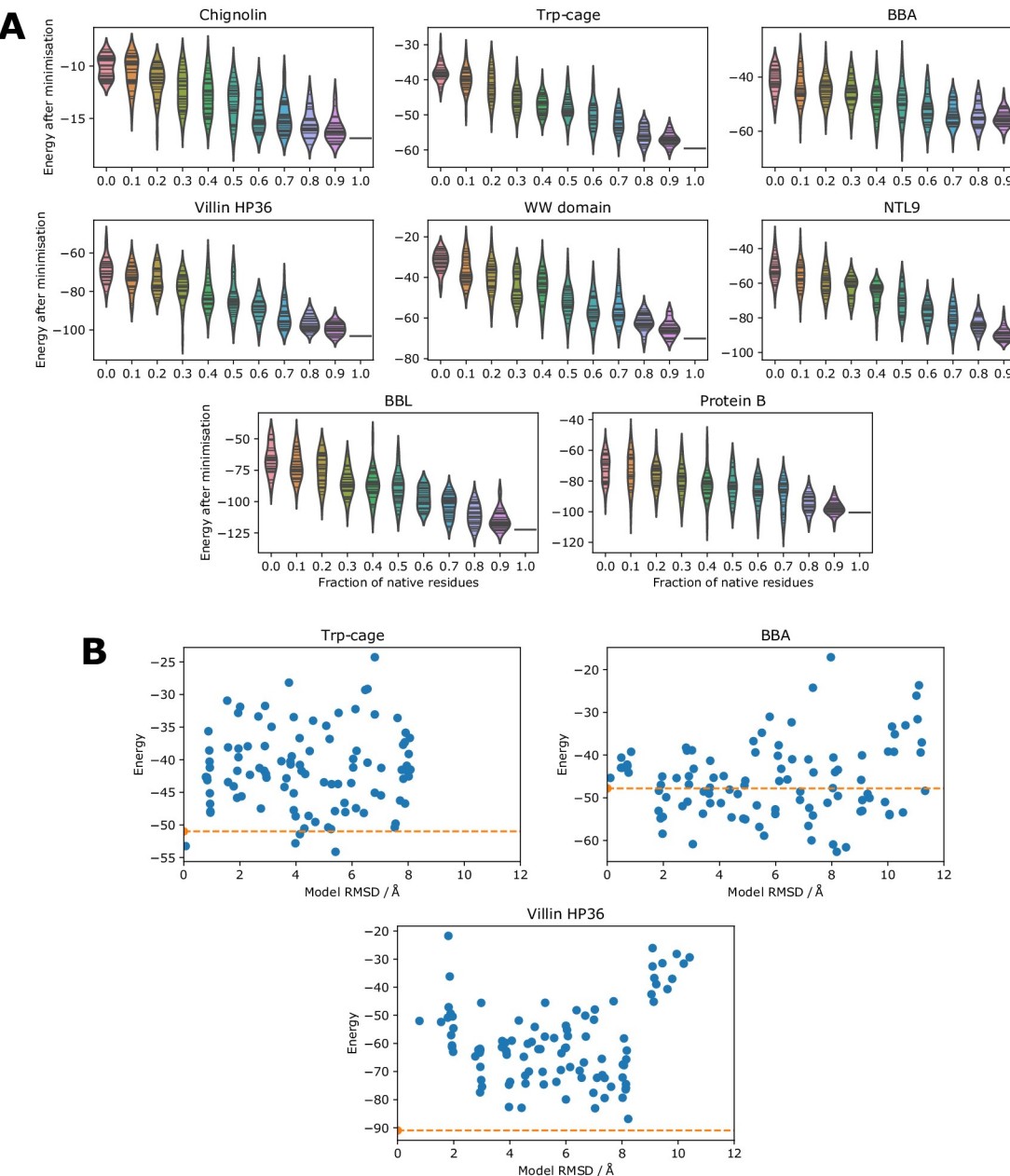

**Fig 5. Protein design and model scoring.** (A) The energy of sequences with varying fractions of native residues threaded onto the native structure. 30 sequences are scored for each fraction. A short minimisation of 100 steps is carried out for each sequence and the energy is recorded at the end of the minimisation. (B) The energy of well-packed decoys generated using 3DRobot is compared to the Cα RMSD of the decoys (blue dots) and the energy of the native structure (orange line). Chignolin was too small to run 3DRobot on.

After 2,000 trials some of the designed sequences match the native sequences, particularly in the core regions with inter-residue interactions. The native fraction for each protein is shown in Table 1 along with analogous results from EvoEF2 [65], which uses an energy function consisting of nine terms and is developed specifically for protein design applications. The results from the potential presented here are comparable to those from EvoEF2, with our potential performing better on the smaller proteins and worse on the larger proteins. Training the

**Table 1. Fixed backbone design.**

| Protein | EvoEF2 fraction of native residues | This work fraction of native residues |
|---|---|---|
| Chignolin | 0.300 | 0.400 |
| Trp-cage | 0.300 | 0.550 |
| BBA | 0.143 | 0.179 |
| Villin HP36 | 0.229 | 0.143 |
| WW domain | 0.303 | 0.152 |
| NTL9 | 0.308 | 0.231 |
| BBL | 0.234 | 0.149 |
| Protein B | 0.106 | 0.085 |

The fraction of residues that match the native residue at the end of the design process is given for this work and for EvoEF2.

potential specifically for protein design and applying it to larger proteins is a topic of future work.

We also investigate whether the potential is able to distinguish between native structures and close decoys. Models up to 12 Å from the native structure were obtained using 3DRobot [66], which generates diverse and well-packed decoys using fragment assembly and energy minimisation. As shown in Fig 5B, the native structure has low energy compared to the decoys for Trp-cage and villin HP36. Chignolin was too small to run 3DRobot on. For BBA many decoys are lower in energy than the native structure. A number of these lower energy decoys are topological mirrors in which the β turn faces the other way, but the structure of the protein is otherwise native-like. The problem of mirror topologies has been discussed previously for protein structure determination [67] and de novo protein structure prediction [68].

## Discussion

The purpose of learning a coarse-grained force field for proteins is to demonstrate that DMS can be used to learn all the parameters from scratch in simple, interpretable force fields. It is notable that running training simulations from the native structure, reaching up to 4 Å RMSD to the native structure over 2,000 steps during training, is sufficient to learn a potential that can fold proteins from an extended chain of secondary structure elements over a few million steps. Whilst the nature of the training may give some bias to globular structures, as is the case with most force fields for proteins, we do notice that simulations at higher temperatures lead to unfolded and variable structures. The same potential can also be used for model scoring, despite the energy not being explicitly used at all beyond force calculation during training. This particular force field may be useful for exploring the conformational space of proteins, discovering folding pathways, assessing flexible or disordered regions of proteins, or predicting structure in combination with co-evolutionary or experimental constraints [69]. Other coarse-grained systems may be immediately amenable to DMS, for example protein aggregation [70]. The real possibility for DMS though lies in applying it to all-atom potentials with a variety of loss functions.

Here we have used RMSD as a simple loss function, but there are a variety of possible loss functions for DMS suitable for other systems. Examples include RMSD over the course of a simulation, radius of gyration, the radial distribution function, the flexibility of a set of atoms during the simulation, the distance between two molecules, supramolecular geometry (e.g. assembly of molecules into fibres), the correlation of different particle velocities, the energy of

a system, the temperature of a system, a measure of phase change, steered molecular dynamics, or some combination of the above. Many of the possible properties are based on static reference structures, thermodynamic observables or chemical knowledge, meaning that expensive trajectory data is not necessarily required. Complex constraints that might be difficult to use in the simulation itself can be targeted via the loss function. Any property that can be computed from the system with a meaningful gradient can be used to guide a learned force field to reproduce desired behaviour. This sets DMS apart from contrastive divergence approaches that train entire force fields by teaching them to distinguish native and non-native ensembles. Possible applications include improving force field accuracy on disordered proteins [8], combining DMS with learned atom typing [25] to study protein-ligand binding, improving torsion angle potentials to balance bonded and non-bonded terms [71], exploring whether multi-body terms can make molecular mechanics potentials more accurate, and developing potentials to promote protein-protein docking. The development of software packages appropriate for DMS such as JAX MD [72], TorchMD [18], DeePMD-kit [73], SchNetPack [74], and DiffTaichi [75], along with the effort to make programming languages such as Julia differentiable by default [76], will assist in the development of DMS.

For DMS to be used to parameterise all-atom molecular mechanics force fields—either from scratch, by tuning existing parameters [8], or by adding new atom types—a few issues need to be addressed. Algorithms such as particle mesh Ewald for long-range electrostatic interactions [77] will have to be implemented in differentiable frameworks. The best form for temperature control during training will have to be considered, as stochasticity will likely affect the gradients. In this study, the thermostat was not used during training. Even the form of the numerical integration will have to be explored, as it is unclear that velocity Verlet integration with a standard time step is best-suited to DMS. In previous work damping of velocities or gradients have been used to prevent exploding gradients [34, 75]. It has also been shown that discrete time steps [75] and large time steps [34] can lead to incorrect gradients. Ensuring continuous potentials and reasonable time steps should prevent these issues, and the standard all-atom potentials—harmonic, cosine, Coulomb and Lennard-Jones—are all continuous. However, cutoffs for short-range forces and neighbour lists will have to be carefully considered. Finding the parameters for a simple water model would be an ideal system to start with [18].

DMS, like deep learning, appears to be sensitive to hyperparameters, and generally appropriate choices for these will have to be discovered. Learned potentials being amenable to physical interpretation helps when investigating such issues, as well as alleviating problems of under-specification and shortcut learning identified for neural networks [78]. For example, increasing the learning rate in this study gives jagged potentials, whereas decreasing it leads to prohibitively slow training.

As with analogous developments in deep learning, a limitation of DMS is the GPU memory required. Most deep learning software frameworks are geared towards reverse-mode AD, in which intermediate results of the forward pass of the network are stored and used during the reverse pass to calculate gradients. The requirement to store intermediate results means that it scales linearly in memory with the number of steps for DMS. By contrast, forward-mode AD does not store intermediate results because the gradients are calculated in tandem with the forward pass. The memory required does not therefore increase with the number of steps, though the number of learned parameters does affect the computation speed. Use of forward-mode AD may provide a way for DMS to use the large number of steps during training that would be required to learn molecular mechanics force fields. Advances in hardware, GPU parallelism and algorithmic techniques such as gradient checkpointing, invertible simulations [79], adjoint sensitivity [29, 30] and offloading compute to the CPU [80] also present solutions to the issue of GPU memory.

Though applications of deep learning in biology have been impressive, the models used have been largely taken off the shelf from other fields. DMS provides a general approach to improving and expanding the force fields that have been crucial for biological understanding. Previous limitations are rapidly being addressed by improvements in hardware, specialist software and the ever-increasing amount of experimental data available. This study shows that DMS can use the techniques and frameworks of neural network training but rely on proven, interpretable functional forms rather than deep neural networks themselves.

## Materials and methods

### Dataset

The PDB [61] was searched for protein chains with 20 to 100 residues, no internal missing residues and resolution 2.5 Å or better. These chains were clustered at 30% sequence identity to reduce redundancy. Proteins homologous to those used in the results were removed from the dataset by eliminating overlap at the same ECOD T-level [81] and removing hits when searching the PDB using BLAST with an E-value of 1.0. The resulting chains were randomly split into a training set of 2,004 chains and a validation set of 200 chains used to monitor training. Single sequence secondary structure prediction was carried out using PSIPRED [82]. Details of the proteins used in the results are given in S1 Table in S1 File. Dataset collection and other aspects of the work were carried out using BioStructures.jl [83] and the Bio.PDB module of Biopython [84].

### Molecular simulation

The system was implemented in PyTorch [57]. A protein is represented in a coarse-grained manner with 4 point particles for each residue corresponding to backbone nitrogen, backbone Cα, backbone carbonyl carbon and the centroid of the sidechain heavy atoms. There is no explicit solvent, no periodic boundary conditions and no neighbour list. The masses are set to 15 for N (includes amide H), 13 for Cα (includes H) and 28 for C (includes carbonyl O). The sidechain mass for each amino acid is the sum of the atom masses in the all-atom sidechain, with glycine set artificially heavier at a mass of 10.

The overall learned potential used to obtain the forces at each time step consists of 3 components, each consisting of many individual potentials. Each individual potential consists of a number of energy values corresponding to 140 bins evenly distributed over a specified distance or angle range. Overall there are 29,360 individual potentials and 4,111,000 learnable parameters.

1. *Pairwise distance potentials*. There are 80 atom types (4 atoms for 20 amino acids) and each pair has a distance potential. In addition there are separate potentials for each atom pair on residues close in sequence with residue separations $i \rightarrow i$ to $i \rightarrow i + 4$, allowing the model to learn local constraints separately from global preferences. Covalent bond interactions are included implicitly in the potentials for atom pairs on the same residue, and the $i \rightarrow i + 1$ potentials in the case of the C-N backbone covalent bond. There are 28,960 pairwise distance potentials in total (3,240 general, 120 same residue, 6,400 × 4 close residues). Each distance potential has 140 bins distributed between 1 Å and 15 Å (0.1 Å width per bin). For the close residue potentials the bins are distributed between 0.7 Å and 14.7 Å (0.1 Å width per bin), and for same residue pairs between 0.7 Å and 5.6 Å (0.035 Å width per bin).

2. *Bond angle potentials*. There are 5 bond angles in the model—3 in the backbone, 2 to the sidechain centroid—and there is a separate potential for each of these for each amino acid.

There are 100 bond angle potentials in total ($5 \times 20$). Each angle potential has 140 bins distributed between 60° and 180° (0.86° width per bin).

3. *Torsion angle potentials*. There are 5 torsion angles in the model—3 in the backbone, 2 to the sidechain centroid—and there is a separate potential for each of these for each amino acid. There are also separate potentials for residues predicted as α-helical, β-sheet or coiled to help in secondary structure formation. There are 300 torsion angle potentials in total ($5 \times 20 \times 3$). Each torsion angle potential has 140 bins distributed between -180° and 180° (2.57° width per bin) with an extra bin on either end to allow the derived force to be periodic.

At each simulation time step the force is calculated as the negative gradient of the potential using the following finite differencing procedure for each individual potential:

1. Calculate the current value of the property, e.g. the distance between two atoms or a bond angle.

2. Find the bin $b_i$ with the closest bin centre to the value, excluding the first and last bins. For distances this means that all distances over the maximum bin distance (e.g. 15 Å) are treated as being in the penultimate bin.

3. Calculate $F = \frac{1}{2}(E(b_{i-1}) - E(b_{i+1}))$, where $E(b_i)$ is the energy of bin $b_i$.

4. Multiply $F$ by the appropriate vector on each atom to apply the force [85].

This is shown in Fig 1C. The sum of the resulting forces on each atom is divided by the atomic masses to get the accelerations. This approach is differentiable since it can be implemented in a vectorised manner in PyTorch. Whilst more sophisticated methods such as a sum of Gaussians or spline fitting could be adopted to obtain the force from the potential, this finite differencing procedure was found to be memory-efficient and effective. It was also found to be more effective than learning force values directly, an approach that does not immediately provide an interpretable potential that can be used to calculate energies.

The coordinates and velocities are updated at every time step using the velocity Verlet integrator. If $\mathbf{x}(t)$ is the coordinates at time $t$, $\mathbf{v}(t)$ is the velocities at time $t$, $\mathbf{a}(t)$ is the accelerations at time $t$, and $\Delta t$ is the time step, then the procedure is:

1. Calculate $\mathbf{x}(t + \Delta t) = \mathbf{x}(t) + \mathbf{v}(t)\Delta t + \frac{1}{2}\mathbf{a}(t)\Delta t^2$.

2. Obtain $\mathbf{a}(t + \Delta t)$ from the learned potential as described above using $\mathbf{x}(t + \Delta t)$.

3. Calculate $\mathbf{v}(t + \Delta t) = \mathbf{v}(t) + \frac{1}{2}(\mathbf{a}(t) + \mathbf{a}(t + \Delta t))\Delta t$.

4. Update $t + \Delta t$ to $t$ and go to step 1.

No thermostat was used during training as it was not found to improve performance. Training therefore takes place in the NVE ensemble, with the caveat that the coarse-grained nature of the system means it does not have a conventional volume. A time step of 0.02 was used for training.

During production runs used to obtain results, which consisted of many more steps than during training, the Andersen thermostat was used to keep temperature constant [86]. Production runs therefore take place in the NVT ensemble. At each step, each atom is given a new velocity with probability equal to the time step divided by a coupling constant. The new velocities are drawn from a normal distribution with mean 0 and standard deviation equal to a temperature value. Simulations were run with temperature 0.015 and coupling constant 25, apart from the (AAQAA)₃ repeat peptide helix formation simulations which were run with

temperature 0.022 and coupling constant 100. A lower time step of 0.004 was used during production runs to ensure stability of the simulations. We did try using Langevin dynamics for production runs but found it did not improve the results.

## Training

During training, simulations are started from the native structure. Starting velocities are drawn from a normal distribution with mean 0 and standard deviation 0.1. At the end of each simulation $R_f$ is calculated as the RMSD across all atoms in the coarse-grained model of the final conformation compared to the native structure using the Kabsch algorithm. $\log(1 + R_f)$ was used as the loss function, which was found to give better performance than using $R_f$. PyTorch records operations during the simulation (the forward pass) in a directed acyclic graph. Once the loss is calculated, the graph is traversed backwards (the backward pass) and the known gradient functions at each step combined using the chain rule to obtain the gradients for each parameter. These gradients are used by the optimiser to modify the potential to reduce the $R_f$ from future simulations. At the start of training all values in the potential are set to zero. We did try starting the potentials from PMFs derived from the PDB but this did not improve results. The Adam optimiser [87] was used with a learning rate of $10^{-4}$. Adam maintains different learning rates for each parameter and gave better results than stochastic gradient descent. Gradients are accumulated for 100 proteins before the optimiser updates the gradients. At epoch 38 the Adam optimiser was reset with a lower learning rate of $5 \times 10^{-5}$.

One epoch of training consists of simulating each protein in the training set in a random order. A batch size of 1 was used due to memory constraints. The number of steps in the simulation was increased over the course of training, starting at 250 for the first epoch and increasing by 250 every 5 epochs to a maximum of 2,000 at epoch 36 and beyond. This allowed the model to access lower $R_f$ values during early epochs to learn basic chemical principles such as steric clashing and covalent bonding. Further increasing the step number was prevented by memory limitations of the GPU (32 GB); during training the memory required scales linearly with the number of steps due to the requirements of storing intermediate computations for reverse mode AD. This limitation is not present for production runs, which use memory constant in the number of steps and typically less than 1 GB. Training was carried out on a NVIDIA Tesla V100 for 45 epochs, which took around 2 months. This can likely be sped up using multiple GPUs. The training and validation $R_f$ values throughout training are shown in S1 Fig in S1 File. The protein folding simulations shown in Fig 4B took around 36 hours for 12m steps, equating to around 10 ms per time step. This was constant across protein sizes tested due to the vectorised operations used. Simulation time is around 3 times slower on the CPU depending on the hardware used.

## Analysis

Throughout this study we analyse only coarse-grained models, however all-atom models can be generated using software such as PULCHRA [88] if required. Alanine dipeptide conformations were generated using PeptideBuilder [89] at φ/ψ intervals of 2˚. The (AAQAA)$_3$ repeat peptide simulation was started from a random conformation where each residue was given a φ angle between -180˚ and -30˚ and a ψ angle between -180˚ and 180˚. α-helical fraction was determined by the fraction of non-terminal residues where the φ angle was between -120˚ and -30˚ and the ψ angle was between -60˚ and 30˚. The α-helical fraction was averaged over a window of 5 snapshots either side of the snapshot in question for ease of visualisation. For simulations starting from predicted secondary structure, a residue was given an α-helical starting

conformation (φ -60˚, ψ -60˚) if predicted α-helical and an extended starting conformation (φ -120˚, ψ 140˚) if predicted β-sheet or coiled.

The representative structures in Fig 4B were found using MDAnalysis [90] and were visualised in PyMOL [91] after being run through PULCHRA. UNRES models were generated on the web server [63] using the parameters for the MREMD structure prediction example in the tutorial: the FF2 force field, extended chain start, secondary structure restraints, Berendsen thermostat with 1.0 coupling to the thermal bath, and 8 replicas exchanging every 1,000 steps with temperatures ranging from 270 K to 340 K in steps of 10 K. The number of steps was increased to the maximum of $10^7$. The top ranked model was used. CABS-fold models were generated on the web server [64] with default de novo parameters including CABS temperature 3.5–1.0. The top ranked model was used. In both cases we use the same single sequence secondary structure prediction as for our method.

The fraction of native sequence results involved threading a sequence onto the native structure. Sequences were chosen by mutating a given fraction of residues to amino acids taken from the background distribution of amino acids in the PDB. 30 sequences are generated for each fraction. Each sidechain centroid is placed at a distance from the Cα atom corresponding to the minimum in the learned potential for that amino acid, along the vector linking the Cα atom and the native sidechain centroid. A brief energy minimisation of 100 steps in the learned potential is carried out and the final energy is used as the energy of the sequence. For the fixed backbone design task 2,000 trial mutations were made. Each trial involved mutating one residue to an amino acid taken from the PDB distribution, calculating the new energy by threading the sequence and running energy minimisation as above, and accepting or rejecting the mutation. Mutations resulting in lower energy are always accepted, and mutations resulting in a higher energy of 10 or more energy units are always rejected. Mutations resulting in an energy increase up to 10 energy units are accepted with a probability that changes linearly from 0.25 at the start of the simulation to 0.0 at 1,000 trials, and remains at 0.0 for the remaining trials.

EvoEF2 [65] fixed backbone design runs were carried out with default de novo design parameters by running 'EvoEF –command = ProteinDesign –monomer –pdb = input.pdb'. 3DRobot models were generated using the 3DRobot standalone software with default parameters [66]. Plots throughout were produced with Matplotlib [92], seaborn [93] and PyEMMA [94] for Fig 3B.

## Supporting information

**S1 File. Contains all the supporting tables and figures.**
(PDF)

## Acknowledgments

We thank Nick Charron and Cecilia Clementi for providing the all-atom alanine dipeptide data from Wang et al. 2019 [15]. We thank the UCL Bioinformatics Group for useful discussions.

## Author Contributions

**Conceptualization:** Joe G. Greener.

**Data curation:** Joe G. Greener.

**Funding acquisition:** David T. Jones.

**Investigation:** Joe G. Greener.

**Methodology:** Joe G. Greener.

**Project administration:** David T. Jones.

**Software:** Joe G. Greener.

**Supervision:** David T. Jones.

**Validation:** Joe G. Greener.

**Writing – original draft:** Joe G. Greener.

**Writing – review & editing:** Joe G. Greener.

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
