## [Decision Letter · Decision Letter 0]

4 May 2021

PONE-D-21-12960

Differentiable molecular simulation can learn all the parameters in a coarse-grained force field for proteins

PLOS ONE

Dear Dr. Greener,

Thank you for submitting your manuscript to PLOS ONE. After careful consideration, we feel that it has merit but does not fully meet PLOS ONE’s publication criteria as it currently stands. Therefore, we invite you to submit a revised version of the manuscript that addresses the points raised during the review process.

We look forward to receiving your revised manuscript.

Kind regards,

Yang Zhang

Academic Editor

PLOS ONE

Journal Requirements:

PLOS requires an ORCID iD for the corresponding author in Editorial Manager on papers submitted after December 6th, 2016. Please ensure that you have an ORCID iD and that it is validated in Editorial Manager. To do this, go to ‘Update my Information’ (in the upper left-hand corner of the main menu), and click on the Fetch/Validate link next to the ORCID field. This will take you to the ORCID site and allow you to create a new iD or authenticate a pre-existing iD in Editorial Manager. Please see the following video for instructions on linking an ORCID iD to your Editorial Manager account: https://www.youtube.com/watch?v=_xcclfuvtxQ

Please include your tables as part of your main manuscript and remove the individual files. Please note that supplementary tables (should remain/ be uploaded) as separate "supporting information" files

Reviewers' comments:

Reviewer's Responses to Questions

**Comments to the Author**

1. Is the manuscript technically sound, and do the data support the conclusions?

Reviewer #1: Yes

Reviewer #2: Yes

2. Has the statistical analysis been performed appropriately and rigorously? 

Reviewer #1: N/A

Reviewer #2: Yes

3. Have the authors made all data underlying the findings in their manuscript fully available?

Reviewer #1: Yes

Reviewer #2: Yes

4. Is the manuscript presented in an intelligible fashion and written in standard English?

Reviewer #1: Yes

Reviewer #2: Yes

5. Review Comments to the Author

Reviewer #1: Greener and Jones proposed to use automatic differentiation adapted from deep learning for learning all the parameters for molecular dynamics force field. They examined this method in a simple coarse-grained force field and used the learned force potential to study protein folding, scoring, and design on a set of small proteins. The method is useful and interesting, and the results seem promising. I have the following comments.

1. They mentioned that the modeling is trained on a set of 2004 diverse proteins up to 100 residues, but in the case studies (Table S1), the largest protein is only containing 35 amino acids. I wonder how their learned potential performs on larger proteins, e.g., ~100 amino acids, for folding, scoring, and design.

2. The test proteins are too few and small. They should include more and larger proteins for testing.

3. In the so-called protein design in this work, the authors did not systematically perform de novo sequence designs on fixed backbones. Instead, they evaluated the energy distributions of a set of sequence decoys with varying fraction of native sequence. Besides, they mutated residues according to PDB distributions. I wonder what is the performance of their learned force field for the computational de novo sequence design benchmark, i.e., “native sequence recapitulation rate”? Again, the authors just compared the native sequence and a set of sequence decoys on a set of mini proteins. This cannot represent a general performance of their useful learned potentials. I want to know 1) the performance of their potential on a larger set of larger proteins and 2) the comparison of their potential on native sequence recapitulation with other protein design approaches such as Rosetta and EvoEF2, just as they did for protein folding to compare with UNRES and CABS-fold.

4. Why NVE for training while NVT for testing? To my knowledge, NPT ensemble is often used for MD.

5. Some Figures are distractedly discussed in the manuscript, e.g., Figure 3B should be Figure 3C, and some figures/subfigures have never been mentioned at all.

Reviewer #2: The paper describes an interesting method for force-field optimization, which is based on machine learning. The Authors developed a new coarse-grained model of proteins, in which each residue is represented by 4 interaction sites (N, carbonyl-C, Calpha, and sidechain center). In optimization steps, whole microcanonical MD simulations are performed on the training proteins, starting from their experimental structures, and the potentials are optimized by using the Adam algoritm with automatic derivative calculations, the target function being log(1+rmsd), where rmsd is the root mean standard deviation from the experimental structure. Whole potential curves are determined; hence the potentials depend only on a single distance or angle/torsional angle, no dependence on orientation included. The optimized potentials were tested only against mini-proteins in de novo folding simulations but this does not seem to be a problem, because the Authors' objective was to demonstrate the principle rather than to produce a force field of practical application at this point. Moreover, the Authors have demonstrated that the potentials perform well in threading with minimization and in inverse folding.

The paper is very interesting and well written. I enjoyed readinig it. However, the following minor points should be addressed before it is accepted for publication:

1. Page 15, the "Training" section. Some more details should be given about the optimization procedure, in particular how te Adam optimizer with automatic derivatives works. Only the description of the calculation of the loss (target) function is provided but how are the gradients of the target function calculated? Referring to PyTorch is not a sufficient description.

2. How were the simulations with CABS-fold and UNRES-server carried out? The Authors state that both servers use secondary-structure prediction but, by default, both run in the ab initio mode. Therefore, the Authors should state that they input the secondary-structure information. Also, the UNRES server supports three force-field variants: the old FF2 (which is the default), OPT-WTFSA-2 [JCIM, 57, 2364-2377 (2017)] and the latest (and most advanced scale-consistent variant [NEWCT-9P (JCP, 150, 155104 (2019)]. I guess that the Authors used the FF2 variant, but this should be stated. Also, UNRES when run in MREMD mode produces 5 clusters of conformations. Did the Authors include the rank#1 structure or the lowest-RMSD structure in the analysis? Besides, UNRES server can also be run in canonical mode and it provides RMSD along the trajectory, from which the distribution can be extracted, which could be compared with those from the learned potential.

3. I am somehow puzzled that so long wall-clock time /10,000,000 steps is required (36 hrs on GPU; page 7, line 14 from the bottom). For 1,000,000 steps with the BBA mini-protein, UNRES server required 900 secs. wall-clock time, which translates to 2,5 wall-clock hrs per 10,000,000 steps on a single INTEL core (no GPU use). From my experience, CABS is comparable in timing or even faster (unfortunately, the CABS server was not functioning properly at the time I was writing this review). Model complicacy seems to comparable; CABS has 3 interaction sites (Calpha, Cbeta SC) and UNRES 2 (peptide groups and SC, but more complicated potentials) and, therefore, some optimization might be missing in force calculation. One thing that could be improved would be to store the numerical derivatives of the potentials in distance in addition to the potentials; this could save one subtraction and one division (point 3 in page 14). Also, symmetric divided differences could be used to improve the accuracy of the forces.

4. The description of the optimization procedure suggests that the experimental structures of the training proteins are only perturbed by running MD in the NVE mode. An immediate concern is that the potentials obtained that way will be biased towards the experimental structures. The fact that there are many training proteins probably makes this concern less serious but the Authors should provide more discussion about the transferability problem (both to non-native states and to other proteins). A maximum-likelihood approach, in which non-native structures are taken into account is described in refs. 37 and 38; also, there are recent papers by the D.E. Shaw group, in which they parameterize the all-atom force field to handle intrinsically-disordered proteins.

5. Figure 4. The superposition of the structure of villing headpiece obtained with the learned potentials does not seem to have the RMSD of 7.38 A. It rather looks like the perturbed native structure with about 2 A RMSD (see the left part of the panel that shows the RMSD distributions). Also, the fact that no conformation obtained in the simulation of villin started from the structure generated with secondary-structure prediction reached 12 low RMSD in 12 million steps raises concern. On the other hand, the simulation started from the experimental structure did not leave the native basin (the lef-bottom panel of Figure 4). If this simulation also lasted 12 million steps, the ergodicity of the simulations is of concern. Perhaps more shorter trajectories should be run.

6. For the reader's benefit, the Authors could mention other approaches at force-field optimization, including those of Crippenn and colleagues and Wolynes and colleagues of the 1990's, as well as the mutiscale coarse-grained force matching method developed by the Voth group.

6. PLOS authors have the option to publish the peer review history of their article (what does this mean?). If published, this will include your full peer review and any attached files.

Reviewer #1: No

Reviewer #2: No

---

## [Author Response · Author response to Decision Letter 0]

24 Jun 2021

See attached file for responses to reviewer comments.

---

## [Decision Letter · Decision Letter 1]

21 Jul 2021

PONE-D-21-12960R1

Differentiable molecular simulation can learn all the parameters in a coarse-grained force field for proteins

PLOS ONE

Dear Dr. Greener,

Thank you for submitting your manuscript to PLOS ONE. After careful consideration, we feel that it has merit but does not fully meet PLOS ONE’s publication criteria as it currently stands. Therefore, we invite you to submit a revised version of the manuscript that addresses the points raised during the review process.

We look forward to receiving your revised manuscript.

Kind regards,

Yang Zhang

Academic Editor

PLOS ONE

Journal Requirements:

Reviewers' comments:

Reviewer's Responses to Questions

**Comments to the Author**

1. If the authors have adequately addressed your comments raised in a previous round of review and you feel that this manuscript is now acceptable for publication, you may indicate that here to bypass the “Comments to the Author” section, enter your conflict of interest statement in the “Confidential to Editor” section, and submit your "Accept" recommendation.

Reviewer #1: All comments have been addressed

Reviewer #2: All comments have been addressed

2. Is the manuscript technically sound, and do the data support the conclusions?

Reviewer #1: Yes

Reviewer #2: Yes

3. Has the statistical analysis been performed appropriately and rigorously? 

Reviewer #1: N/A

Reviewer #2: Yes

4. Have the authors made all data underlying the findings in their manuscript fully available?

Reviewer #1: Yes

Reviewer #2: Yes

5. Is the manuscript presented in an intelligible fashion and written in standard English?

Reviewer #1: Yes

Reviewer #2: Yes

6. Review Comments to the Author

Reviewer #1: (No Response)

Reviewer #2: The Authors have addressed all of my comments and I only have a minor suggestion: in page 17, lines 15-16 from the bottom the Authors state that they used Berendsen thermostat and the 0.02 Langevin scaling when running simulations on the UNRES server. The server uses either the Berendsen or the Langevin thermostat and if the Berendsen thermostat is specified, any Langevin thermostat settings are ignored, becuse the Langevin thermostat is not used. Therefore, unelss the Authors ran part of UNRES server simulations with the Berendsen and part with the Langevin thermostat (it should be stated in the manuscript, if so), they should delete the mention of the Langevin scaling factor.

7. PLOS authors have the option to publish the peer review history of their article (what does this mean?). If published, this will include your full peer review and any attached files.

Reviewer #1: No

Reviewer #2: No

---

## [Author Response · Author response to Decision Letter 1]

28 Jul 2021

We have made the minor change as requested by reviewer 2 by removing the mention of the unused Langevin scaling factor. We believe the manuscript is now ready for publication.

---

## [Editor Report · Decision Letter 2]

20 Aug 2021

Differentiable molecular simulation can learn all the parameters in a coarse-grained force field for proteins

PONE-D-21-12960R2

Dear Dr. Greener,

We’re pleased to inform you that your manuscript has been judged scientifically suitable for publication and will be formally accepted for publication once it meets all outstanding technical requirements.

Kind regards,

Yang Zhang

Academic Editor

PLOS ONE
---

## [Editor Report · Acceptance letter]

25 Aug 2021

PONE-D-21-12960R2 

Differentiable molecular simulation can learn all the parameters in a coarse-grained force field for proteins 

Dear Dr. Greener:

I'm pleased to inform you that your manuscript has been deemed suitable for publication in PLOS ONE. Congratulations! Your manuscript is now with our production department. 

Kind regards, 

on behalf of

Dr. Yang Zhang 

Academic Editor

PLOS ONE